# Patients with Cardiac Implantable Electronic Device Undergoing Radiation Therapy: Insights from a Ten-Year Tertiary Center Experience

**DOI:** 10.3390/jcm11174990

**Published:** 2022-08-25

**Authors:** Simone Gulletta, Giulio Falasconi, Lorenzo Cianfanelli, Alice Centola, Gabriele Paglino, Manuela Cireddu, Andrea Radinovic, Giuseppe D’Angelo, Alessandra Marzi, Simone Sala, Nicolai Fierro, Caterina Bisceglia, Giovanni Peretto, Nadia Di Muzio, Paolo Della Bella, Pasquale Vergara, Italo Dell’Oca

**Affiliations:** 1Department of Arrhythmology and Cardiac Electrophysiology, San Raffaele Hospital, Via Olgettina 60, 20132 Milan, Italy; 2Centro Cardiologico Monzino, Via Carlo Parea 4, 20138 Milan, Italy

**Keywords:** radiotherapy, cardiac implantable electronic device, pacemaker, cancer, implantable cardioverter defibrillator

## Abstract

Background: The number of patients with cardiac implantable electronic devices (CIEDs) receiving radiotherapy (RT) is increasing. The management of CIED-carriers undergoing RT is challenging and requires a collaborative multidisciplinary approach. Aim: The aim of the study is to report the real-world, ten-year experience of a tertiary multidisciplinary teaching hospital. Methods: We conducted an observational, real-world, retrospective, single-center study, enrolling all CIED-carriers who underwent RT at the San Raffaele University Hospital, between June 2010 and December 2021. All devices were MRI-conditional. The devices were programmed to an asynchronous pacing mode for patients who had an intrinsic heart rate of less than 40 beats per minute. An inhibited pacing mode was used for all other patients. All tachyarrhythmia device functions were temporarily disabled. After each RT session, the CIED were reprogrammed to the original settings. Outcomes included adverse events and changes in the variables that indicate lead and device functions. Results: Between June 2010 and December 2021, 107 patients were enrolled, among which 63 (58.9%) were pacemaker carriers and 44 (41.1%) were ICD carriers. Patients were subjected to a mean of 16.4 (±10.7) RT sessions. The most represented tumors in our cohort were prostate cancer (12; 11%), breast cancer (10; 9%) and lung cancer (28; 26%). No statistically significant changes in device parameters were recorded before and after radiotherapy. Generator failures, power-on resets, changes in pacing threshold or sensing requiring system revision or programming changes, battery depletions, pacing inhibitions and inappropriate therapies did not occur in our cohort of patients during a ten-year time span period. Atrial arrhythmias were recorded during RT session in 14 patients (13.1%) and ventricular arrhythmias were observed at device interrogation in 10 patients (9.9%). Conclusions: Changes in device parameters and arrhythmia occurrence were infrequent, and none resulted in a clinically significant adverse event.

## 1. Introduction

Radiotherapy (RT) is the cornerstone for the treatment of various types of cancer. Up to 50% of malignancies require RT for either curative or palliative intent. Because of shared risk factors, a substantial number of cancer patients also have pre-existing cardiovascular disease when starting cancer treatment. The number of patients with cardiac implantable electronic devices (CIEDs) receiving RT is therefore increasing [1]. There are two main categories of CIED: pacemakers (PM) and implantable cardiac defibrillators (ICD). During treatment planning procedures, physicians must consider the radiation dose involving organs-at-risk (OAR), defined as healthy organ tissues close to the irradiated target; thus, CIEDs could be considered as OARs during RT planning.

In all patients with CIED, RT can affect the correct functioning of the devices. This can happen with three described mechanisms: through direct ionizing radiation, through electromagnetic interference and through scattered radiation. All of these can lead to transient or permanent malfunctions. According to previously published literature, device malfunctions can occur in as high as 3% of RT treatments, posing a substantial issue in clinical practice [2]. When a patient has a CIED, RT planning should ensure that no radiation beam is directed at or through the device to minimize the absorbed dose. Another important issue during RT is the occurrence of nuclear reactions within the linear accelerator causing neutron contamination. This event is especially harmful to CIEDs, since scatter radiation can cause malfunctioning of the device even from long distance to the target. This occurs with increasing energy photons (>10 MV) and electrons (>20 MeV) or with proton therapy [3].

The management of CIED-carriers undergoing RT is challenging and requires a collaborative multidisciplinary approach [4]. In 2018, a multidisciplinary working group redacted an expert consensus with the aim of providing national guidance for clinicians, therapeutic radiographers and medical physicists on the management of CIED-carriers undergoing RT [4]. According to the consensus, the risk of CIED malfunctions depends on the RT site, modality and energy, and patient conditions associated with potential risk, such as PM dependency and ICD-carriers. These features allow clinicians to divide patients in three risk categories: low, moderate, and high risk of CIED malfunctions. The authors proposed a personalized management of CIED-carriers based on the proposed risk stratification.

The goals of this study were as follows: (i) to determine the incidence of CIED malfunction; (ii) to characterize the various types of malfunctions that occur; and (iii) to report the real-world, ten-year experience of a tertiary multidisciplinary teaching hospital.

## 2. Methods

We conducted an observational, real-world, retrospective, single-center study; we retrospectively enrolled all CIED-carriers who underwent RT at the San Raffaele University Hospital, between June 2010 and December 2021. The study complies with the Declaration of Helsinki and was approved by the Institutional Ethics Committee.

### 2.1. Patient Sample

All enrolled patients were carriers of PM or ICD, which had been previously implanted according to the current guidelines [5], and had clinical indications for RT. No patients with recent implantation within 4 weeks, with epicardial leads or with permanent non-functional leads had undergone RT sessions. No patients were excluded for clinical characteristics.

### 2.2. CIED Characteristics

Regarding CIED information, the baseline data collected included the type of the device (PM, ICD), device manufacturer, device location, the number of leads, and pacing dependency, defined as the absence of spontaneous rhythm > 30 bpm. All devices were MRI-conditional.

### 2.3. Oncological Characteristics, Radiotherapy and Radiological Exposure

With regards to the radiotherapy treatment, the most common techniques for irradiation were the Intensity-Modulated Radiation Therapy (IMRT) techniques, mainly by means of helical tomotherapy. In addition, 3D conformal techniques or stereotactic ablative body radiotherapy were also used. Information on the anatomical region that was irradiated and the number of fractions were also collected. Moreover, the RT data included start date, end date, type of primary tumor, beam type and beam energy, number of fractions, the total maximal (Dmax) and mean (Dmean) radiation dose delivered to the CIED, the cumulative tumor dose, and the fraction dose. For the purpose of this analysis, beam energies were categorized as neutron-producing (photon beam energy > 10 MV) and non-neutron-producing (photon beam energy ≤ 10 MV and electron beam energy ≤ 18 MeV). For clinical purposes, we divided the radiation exposure into 3 zones: zone 1 including the head and the neck, zone 2 including the chest and pectoral region, and zone 3 including the subdiaphragmatic regions.

### 2.4. Device Programming

Each RT session was supervised either by a physician experienced in CIED programming or by an electrophysiologist. ECG monitoring was recommended for all patients. Devices were programmed to an asynchronous pacing mode for patients who had an intrinsic heart rate of less than 40 beats per minute. An inhibited pacing mode was used for all other patients. All tachyarrhythmia device functions were temporarily disabled. After each RT session, the CIED were reprogrammed to the original settings [5].

### 2.5. Post-Radiotherapy Outcomes

Procedural outcomes included all of the potential adverse events. Prespecified adverse events included generator failure, power-on reset, changes in pacing threshold or sensing that requires system revision or programming changes, battery depletion, pacing inhibition, inappropriate therapies, and cardiac arrhythmias. Immediately before and immediately after the RT, we collected CIED information for all enrolled patients: acquired data included battery voltage and estimated duration; atrial, right ventricle (RV) and left ventricle (LV) capture thresholds; P-wave amplitude, RV and LV R-wave amplitude; and atrial, RV and LV lead impedance.

### 2.6. Statistical Analysis

A structured pre-specified dataset of variables was defined and used to collect patient data. The Shapiro–Wilk test was used to assess the normality of distribution of each variable. Continuous variables are presented as either mean ± standard deviation or median (interquartile range) as appropriate, while categorical variables are presented as frequency distribution and percentage. The Mann–Whitney test and Fisher’s exact test were used to compare continuous and categorical variables, respectively. A level of *p* < 0.05 was chosen for statistical significance. Data were analyzed with R version 3.6.2 software (R Foundation for Statistical Computing, Vienna, Austria).

## 3. Results

### 3.1. Baseline Characteristics

Between June 2010 and December 2021, a total of 107 patients were enrolled, among which 63 (58.9%) were PM carriers and 44 (41.1%) ICD carriers. In the whole study cohort, a total of 11 patients were carriers of CIED with cardiac resynchronization therapy (CRT) function. More specifically, 10 patients in the ICD group had a CRT-D device, and 1 patient in the PM group had a CRT-P device.

Table 1 shows the baseline characteristics concerning patients’ devices and radiotherapy data. Patients were subjected to a mean of 16.4 (±10.7) RT sessions. The number of sessions were similar in both pacemaker and ICD carriers with no statistically significant difference (17.2 ± 10.6 versus 15.1 ± 11.0; *p*-value = 0.38). The RT total mean dose was 46.4 (±15.5) Gy. The device generator received a maximum and a mean dose of 2.8 (±3.8) and 1.0 (±1.3), respectively. The leads received a maximum and a mean dose of 22.5 (±18.8) and 5.4 (±6.5), respectively.

Figure 1 shows the organ location of the tumor (panel B) and the proportion of body areas targeted for the radiotherapy session (panel A). The most represented tumors in our cohort were prostate cancer (12; 11%), breast cancer (10; 9%) and lung cancer (28; 26%), while 58 patients (54%) presented with other types of tumors. The target areas for the radiotherapy session were mainly chest (48; 45%), pelvis (21; 20%) and abdomen (21; 19%). Only a minority underwent head and neck and whole-body radiotherapy, in 15 (14%) and 2 cases (2%), respectively.

In the whole cohort, 25 patients received RT directly on the device, including RT for cancer of the left breast, the left lung, and the mediastinum. In none of these cases the device was explanted and reimplanted in the opposite site.

### 3.2. Changes in Device Parameters

Table 2 and Figure 2 show the device parameter changes before and after the radiotherapy sessions. No statistically significant difference for each single parameter existed before and after radiotherapy in the overall population. Moreover, no statistically significant differences existed before and after radiotherapy in pacemaker and ICD subgroups (Table 3).

### 3.3. Post-Radiotherapy Outcomes

Table 4 shows post-radiotherapy outcomes. Generator failures, power-on resets, changes in pacing threshold or sensing requiring system revision or programming changes, battery depletions, pacing inhibitions and inappropriate therapies did not occur in our cohort of patients during a ten-year time span period.

Atrial arrhythmias were recorded during the RT session in 14 patients (13.1%) with no differences between pacemaker and ICD carriers (15.9% versus 9.1%; *p*-value = 0.39).

Similarly, ventricular arrhythmias during the RT session were observed at device interrogation in 9.9% of patients, with no statistically significant difference in the two groups (8.5% versus 11.9%; *p*-value = 0.74).

None of the arrhythmias recorded during post-RT checks are due to oversensing or CIED malfunctioning.

## 4. Discussion

Our study demonstrated that RT can be performed safely in CIED carriers, independently of the type of the device (both in PM and in ICD carriers) and of the type of the radiation delivered. The absence of major adverse events, such as inappropriate shock delivery or device permanent malfunctions, demonstrated that RT has no immediate side effects.

In order to prevent RT-induced CIED malfunctions, it is important to follow an algorithm of risk assessment, such as to classify patients at low, medium or high risk. This stratification allows for the identification of patients’ risk before RT, and to manage the CIEDs in a correct way during and after the RT sessions.

As mentioned before, the potential sources of CIED malfunctions caused by RT are the production of ionizing radiations, the electromagnetic interference, and the production of scatter radiations [6]. The first mechanism can interact with semiconductor components in the electrical circuit of the contemporary CIEDs by loading the silicon dioxide insulator with an excess of electron-hole pairs. The consequence is that the net positive charge that is accumulated can alter the voltage threshold of the device. The second malfunction mechanism, the electromagnetic interference, can lead to signal disturbances resulting in altered sensing or stimulation. In literature, a rare case is reported in which a patient undergoing RT for a tumor in the neck area received an inappropriate shock from the ICD, following an incorrect ventricular fibrillation detection because of T-wave oversensing [7]. Another effect of electromagnetic interference can be the full reset of the device or, in the worst-case scenario, the complete device failure, which can be a life-threatening condition in pacemaker-dependent patients. The last described mechanism of CIED malfunction is the production of scatter radiations with neutron production. This occurs with increasing energy photons (>10 MV) and electrons (>20 MeV) or with proton therapy [3]. This neutron production mechanism is described as the major predictor of contemporary CIED malfunctions, because of nuclear reactions that can damage CIEDs from a high distance from the target organ.

In accordance with our results, in the most recent studies it is demonstrated that neither the radiation dose nor device distance from the organ-at-risk correlates with device malfunctions [8]. In fact, nuclear reactions, which represent the most frequent mechanism involved in device malfunctions, can play a role independently from the radiation dose (since nuclear reactions can occur even at low doses when some types of energies are produced) and from distance to the target organ (because of the production of scatter radiations). Based on these evidences, the solutions indicated in the past to minimize the total absorbed dose from the device could be of uncertain utility. The first proposed solution was surgical device relocation before RT. This option is suitable only under certain circumstances: in case of patients with low surgical risk; in case of patients who do not require chemotherapy as this could increase the risk of infection; or in case the original location of the device could lessen the efficacy of RT delivery to the target organ. Another proposed mechanism was the use of lead shields to minimize device exposure to ionizing radiations. Nonetheless, even though it can be safe to avoid electromagnetic interference, lead shields do not protect CIEDs from scatter radiations. On the contrary, they increase the risk of device malfunctions because of neutrons production from beam collision.

In general terms, it is important to calculate the theoretical absorbed dose to the device, in order to be able to predict potential malfunctions before starting RT. The absorbed dose will be greater if the planned target of radiotherapy includes neck, thorax, and the upper extremities (with a threshold of 2 Gy, as mentioned in the first guidelines in 1994) [9]. Moreover, the absorbed dose will be higher with certain types of energies, such as neutron production energies that are the most dangerous.

After RT planning, a complete CIED evaluation should be made before initiation of RT. The treatment team should be informed of the type of the device (PM or ICD), the extent of PM dependency, the minimum programmed pacing rate and the maximum programmed tracking and sensor rates [6]. The characteristics of RT sessions, combined with complete CIED evaluation, allow physicians to do an appropriate risk assessment, classifying patients at low, intermediate, and high risk [9,10,11]. Many clinical practice guidelines have been published over the years, but a common ground can be found. For example, according to 2018 guidelines [4], the patient is automatically classified at high risk if expected to receive more than 10 Gy, if he/she is PM-dependent, or in case of frequent ICD interventions in the past. This risk stratification allows to estimate the probability of adverse events related to device malfunctions. For this reason, it is important to reprogram devices before beginning RT. In our study, we set in an asynchronous pace mode for PM-dependent patients (e.g., with an intrinsic heart rate less than 40 bpm), we disabled temporarily tachyarrhythmia device functions, and we set all other patients on an inhibited pace mode. Devices were reported to the original settings after each RT session. This approach can explain why some episodes of ventricular or atrial arrythmias occurred, but without evidence of clinical manifestations since they were detected only at device interrogation. Importantly, CIED interrogations showed no significant difference in all of the parameters checked in the control after RT sessions. In conclusion, this single tertiary center ten-year experience demonstrated that RT is safe in CIEDs carriers because no device malfunction was noticed, even if patients who underwent RT varied from a wide range of absorbed dose and radiation beam energy.

## 5. Study Limitations

Our study has some limitations: (i) we report on an observational retrospective study which could be affected by biases (including lack of uniformity of the medical indications used) that cannot be fully adjusted for; (ii) the data were acquired at a single center and may not be generalizable to other RT facilities; (iii) defibrillation threshold testing post-RT was not performed for ICD devices; (iv) we have not collected long-term follow-up data; (v) and finally, device technology is constantly in evolution and interference with future systems cannot be ruled out.

## 6. Conclusions

We studied the safety of RT performed in 107 patients with MRI-conditional PM and ICD. Changes in device parameters and arrhythmia occurrence were infrequent, and none resulted in a clinically significant adverse event.

## Figures and Tables

**Figure 1 jcm-11-04990-f001:**
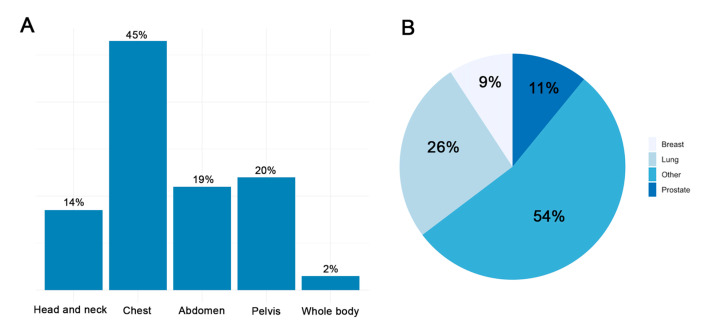
The proportion of body areas (**A**) and location of the tumor (**B**) treated with radiation therapy.

**Figure 2 jcm-11-04990-f002:**
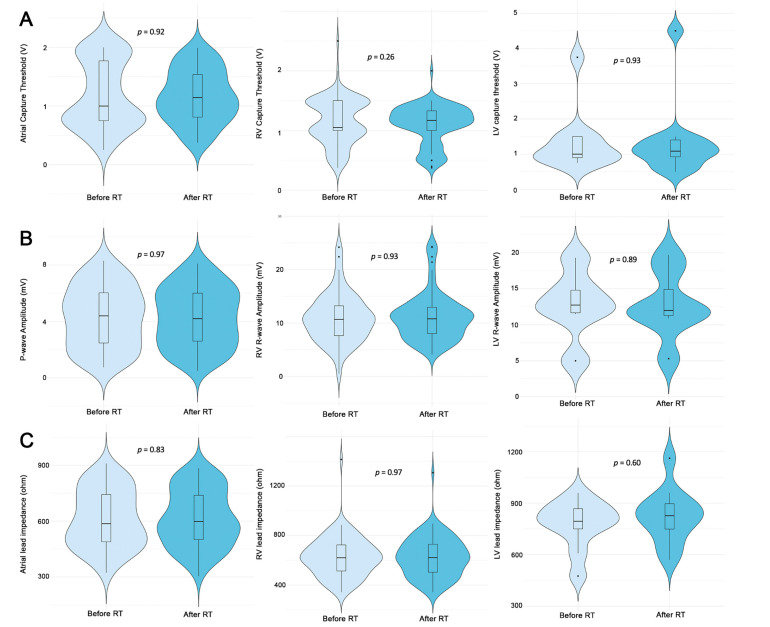
Changes in device parameters: capture thresholds (**A**), sensing amplitude (**B**), and leads impedance (**C**).

**Table 1 jcm-11-04990-t001:** Patients’ devices and radiotherapy data.

	Total Patients (n = 107)	Pacemaker Carriers (n = 63)	Implantable Cardioverter Defibrillator Carriers (n = 44)	*p* Value
Age (years)	75.8 ± 7.0	77.4 ± 7.5	74.2 ± 6.3	0.07
Manufacturer Medtronic Biotronik St. Jude/Abbott Boston Sorin				
	45 (42.1)	31 (49.2)	14 (31.8)	
	16 (15.0)	7 (11.1)	9 (20.5)	
	26 (24.3)	10 (15.9)	16 (36.4)	
	15 (14.0)	11 (17.5)	4 (9.1)	
	5 (4.7)	4 (6.3)	1 (2.3)	
RT sessions	16.4 ± 10.7	17.2 ± 10.6	15.1 ± 11.0	0.38
RT total dose (Gy)	46.4 ± 15.5	46.9 ± 15.4	45.7 ± 15.7	0.79
RT fractions	16.0 ± 10.3	16.3 ± 10.7	15.7 ± 9.9	0.87
Device Maximum Dose (Gy)	2.8 ± 3.8	3.0 ± 4.2	2.6 ± 3.1	0.83
Device Mean Dose (Gy)	1.0 ± 1.3	0.9 ± 1.1	1.0 ± 1.6	0.69
Lead Maximum Dose (Gy)	22.5 ± 18.8	22.5 ± 18.8	22.5 ± 19.2	0.89
Lead Mean Dose (Gy)	5.4 ± 6.5	5.7 ± 6.5	4.9 ± 6.5	0.30

Results are reported as n (%) for categorical variables and mean ± standard deviation for continuous variables. RT = Radiotherapy.

**Table 2 jcm-11-04990-t002:** Changes in device parameters.

	Before Radiotherapy (n = 107)	After Radiotherapy (n = 107)	*p* Value
Atrial capture threshold *	1.2 ± 0.6	1.2 ± 0.5	0.92
P-wave amplitude	4.3 ± 2.2	4.3 ± 2.1	0.97
Atrial lead impedance	611.5 ± 155.2	614.6 ± 152.5	0.83
RV capture threshold *	1.2 ± 0.4	1.1 ± 0.3	0.26
RV R-wave amplitude	11.0 ± 4.6	11.2 ± 4.5	0.93
RV lead impedance	623.3 ± 158.1	622.8 ± 158.1	0.97
LV capture threshold *	1.3 ± 0.9	1.3 ± 1.0	0.93
LV R-wave amplitude	12.8 ± 4.7	12.8 ± 4.7	0.89
LV lead impedance	779.9 ± 138.0	824.4 ± 155.5	0.60

Results are reported as mean ± standard deviation for continuous variables. LV = Left Ventricle; RV = Right Ventricle. * Atrial, RV and LV capture thresholds were measured with the same pulse width.

**Table 3 jcm-11-04990-t003:** Changes in device parameters according to device type.

Pacemakers (n = 63)			
	Before Radiotherapy (n = 63)	After Radiotherapy (n = 63)	*p* Value
Atrial capture threshold *	1.2 ± 0.4	1.2 ± 0.6	0.94
P-wave amplitude	4.5 ± 2.0	4.6 ± 2.0	0.80
Atrial lead impedance	622.6 ± 146.3	619.5 ± 147.6	0.87
RV capture threshold *	1.1 ± 0.3	1.1 ± 0.4	0.90
RV R-wave amplitude	10.7 ± 3.6	10.7 ± 4.3	0.92
RV lead impedance	644.1 ± 174.6	646.0 ± 179.4	0.94
LV capture threshold *	4.5 ± 0.9	3.8 ± 1	0.32
LV R-wave amplitude	12.8 ± 4.7	12.8 ± 4.7	0.89
LV lead impedance	570.0 ± 138.5	608.0 ± 155.5	0.32
**Implantable Cardioverter Defibrillator (n = 44)**			
	**Before Radiotherapy** **(n = 44)**	**After Radiotherapy** **(n = 44)**	***p* Value**
Atrial capture threshold *	1.2 ± 0.5	1.2 ± 0.5	0.92
P-wave amplitude	4.0 ± 2.2	4.0 ± 2.3	0.75
Atrial lead impedance	603.6 ± 161.9	600.8 ± 166.1	0.88
RV capture threshold *	1.1 ± 0.3	1.2 ± 0.4	0.09
RV R-wave amplitude	11.8 ± 5.3	11.4 ± 5.0	0.88
RV lead impedance	593.3 ± 128.1	591.8 ± 117.5	0.97
LV capture threshold *	1.0 ± 0.3	1.1 ± 0.3	0.91
LV R-wave amplitude	12.8 ± 4.7	12.8 ± 4.7	0.89
LV lead impedance	847.5 ± 139.8	797.1 ± 132.5	0.60

Results are reported as mean ± standard deviation for continuous variables. LV = Left Ventricle; RV = Right Ventricle. * Atrial, RV and LV capture thresholds were measured with the same pulse width.

**Table 4 jcm-11-04990-t004:** Post-radiotherapy outcomes.

	Total Patients (n = 107)	Pacemaker Carriers (n = 63)	Implantable Cardioverter Defibrillator Carriers (n = 44)	*p* Value
Generator failures	0 (0)	0 (0)	0 (0)	0.99
Power-on resets	0 (0)	0 (0)	0 (0)	0.99
Changes in pacing threshold requiring system revision or programming changes	0 (0)	0 (0)	0 (0)	0.99
Changes in sensing threshold requiring system revision or programming changes	0 (0)	0 (0)	0 (0)	0.99
Battery depletions	0 (0)	0 (0)	0 (0)	0.99
Pacing inhibitions	0 (0)	0 (0)	0 (0)	0.99
Inappropriate therapies	0 (0)	0 (0)	0 (0)	0.99
Atrial Arrhythmias during RT session period	14 (13.1)	10 (15.9)	4 (9.1)	0.39
Ventricular Arrhythmias during RT session period	10 (9.9)	5 (8.5)	5 (11.9)	0.74

Results are reported as mean ± standard deviation for continuous variables and as absolute number (percentage of the total) for categorical variables. RT: radiotherapy.

## Data Availability

The data that support the findings of this study are available from the corresponding author, upon reasonable request.

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
