# Peer review of "Patients with Cardiac Implantable Electronic Device Undergoing Radiation Therapy: Insights from a Ten-Year Tertiary Center Experience"

_jcm, 2022, doi:10.3390/jcm11174990_

Round 1

Reviewer 1 Report

The manuscript by Simone Gulletta et al. aimed to report the real-world ten-year experience of a tertiary multidisciplinary teaching hospital. It was an observational, real-world, retrospective, single-center study. 107 patients were enrolled: 63 with pacemakers and 44 with ICDs. Patients were subjected to a mean of 33 RT sessions. No statistically significant changes in device parameters were recorded before and after radiotherapy. Generator failures, power-on resets, changes in pacing threshold or sensing requiring system revision or programming changes, battery depletions, and inappropriate therapies did not occur in patients. Some atrial and ventricular arrhythmias occurred. The authors concluded that changes in device parameters and arrhythmia occurrence were infrequent, and none resulted in significant adverse events. The study is interesting and showed clinically relevant data but there are some questions:

1. How many patients had CRT?

2. How many patients had radiotherapy directly on the device for example females with breast cancer that needs radiotherapy directly on the device? Were the device explanted in such situations and implanted on the opposite side?

  1.  

  1.  
      1.  

Author Response

Reviewer: 1

The manuscript by Simone Gulletta et al. aimed to report the real-world ten-year experience of a tertiary multidisciplinary teaching hospital. It was an observational, real-world, retrospective, single-center study. 107 patients were enrolled: 63 with pacemakers and 44 with ICDs. Patients were subjected to a mean of 33 RT sessions. No statistically significant changes in device parameters were recorded before and after radiotherapy. Generator failures, power-on resets, changes in pacing threshold or sensing requiring system revision or programming changes, battery depletions, and inappropriate therapies did not occur in patients. Some atrial and ventricular arrhythmias occurred. The authors concluded that changes in device parameters and arrhythmia occurrence were infrequent, and none resulted in significant adverse events. The study is interesting and showed clinically relevant data but there are some questions:

A.: Thanks to reviewer for these positive comments.

  1. How many patients had CRT?

A.: We thank the reviewer for this insightful comment. In the whole study cohort, a total of 11 patients were carriers of CIED with cardiac resynchronization therapy (CRT) function; more specifically, 10 patients in the ICD group had a CRT-D device, and 1 patient in the PM group had a CRT-P device. The following statement has been added in the “Baseline characteristics” paragraph of Results (page 3, lines 138-141):In the whole study cohort, a total of 11 patients were carriers of CIED with cardiac resynchronization therapy (CRT) function; more specifically, 10 patients in the ICD group had a CRT-D device, and 1 patient in the PM group had a CRT-P device.”.

  1. How many patients had radiotherapy directly on the device for example females with breast cancer that needs radiotherapy directly on the device? Were the device explanted in such situations and implanted on the opposite side?

A.: We thank the reviewer for the comment. In the whole cohort, 25 patients received RT directly on the device, including RT for cancer of the left breast, the left lung, and the mediastinum. In none of these cases the device was explanted and reimplanted in the opposite site. The following statement has been added in the “Baseline characteristics” paragraph of Results (page 4, lines 160-162):In the whole cohort, 25 patients received RT directly on the device, including RT for cancer of the left breast, the left lung, and the mediastinum. In none of these cases the device was explanted and reimplanted in the opposite site.”.

Reviewer 2 Report

The authors presented an important rewiew of clinical significance regarding Radiation therapy with CIED patients.

This seems to be n very informative paper that demonstrates the safety of the guideline's recommendation that CIDE patients should avoid RT, especially when treating the chest, with actual data.

One thing about the result site on page 7, it is mentioned that Atrial arrythmia and Ventricular arrythmia were observed in several patients when checked after RT was completed, I would like to know if this was due to oversensing or if it acctually occured.

Author Response

Reviewer: 2

The authors presented an important rewiew of clinical significance regarding Radiation therapy with CIED patients.

This seems to be n very informative paper that demonstrates the safety of the guideline's recommendation that CIDE patients should avoid RT, especially when treating the chest, with actual data.

A.: Thanks to the reviewer for these positive comments.

One thing about the result site on page 7, it is mentioned that Atrial arrythmia and Ventricular arrythmia were observed in several patients when checked after RT was completed, I would like to know if this was due to oversensing or if it acctually occured.

A.: We thank the reviewer for this insightful comment. However, none of the arrhythmias recorded during post RT checks are due to oversensing or CIED malfunctioning. The following statement has been added in the “Post-radiotherapy outcomes” paragraph of Results (page 7, lines 191-192):None of the arrhythmias recorded during post RT checks are due to oversensing or CIED malfunctioning.”.